# Preparation of PP/2D-Nanosheet Composites Using MoS$_2$/MgCl$_2$- and BN/MgCl$_2$-Bisupported Ziegler–Natta Catalysts

**He-xin Zhang [1], Byeong-Gwang Shin [2], Dong-Eun Lee [3],\* and Keun-Byoung Yoon [2],\***

[1] School of Chemistry & Chemical Engineering, Anhui University of Technology, Ma'anshan 143032, China; hxzhang@ahut.edu.cn

[2] Department of Polymer Science and Engineering, Kyungpook National University, Daegu 41566, Korea; bshin@naver.com

[3] School of Architecture & Civil Engineering, Kyungpook National University, Daegu 41566, Korea

\* Correspondence: dolee@knu.ac.kr (D.-E.L.); kbyoon@knu.ac.kr (K.-B.Y.); Tel.: +82-53-950-7540 (D.-E.L.); +82-53-950-7588 (K.-B.Y.)

**Abstract:** Polypropylene/molybdenum disulfied (PP/MoS$_2$) and Polypropylene/hexagonal boron nitride (PP/hBN) nanocomposites with varying concentration (0–6 wt %) were fabricated via in situ polymerization using two-dimensional (2D)-nanosheet/MgCl$_2$-supported Ti-based Ziegler–Natta catalysts, which was prepared through a novel coagglomeration method. For catalyst preparation and interfacial interaction, MoS$_2$ and hBN were modified with octadecylamine (ODA) and octyltriethoxysilane (OTES), respectively. Compared with those of pristine PP, thermal stability of composites was 70 °C higher and also tensile strength and Young's modulus of the composites were up to 35% and 60% higher (even at small filler contents), respectively. The alkyl-modified 2D nanofillers were characterized by strong interfacial interactions between the nanofiller and the polymer matrix. The coagglomeration method employed in this work allows easy introduction and content manipulation of various 2D-nanosheets for the preparation of 2D-nanosheet/MgCl$_2$-supported Ti-based Ziegler–Natta catalysts.

**Keywords:** boron nitride; MoS$_2$; bisupported Ziegler–Natta catalysts; nanocomposites

## 1. Introduction

Layered two-dimensional (2D) nanosheet-based polymer composites have garnered considerable interest in the polymer industry due to their multifunctional properties including low density, high aspect ratio and specific surface area, excellent electrical/thermal conductivity, gas barrier properties, high permittivity, excellent mechanical properties including toughness and compressive strength, etc. [1–4] The graphene-based polyolefin nanocomposites are the most studied in the field of nanosheet-based polymer nanocomposites, owing to their significant potential and excellent physical properties [5–7].

Among the 2D nanosheets, molybdenum disulfide (MoS$_2$) and hexagonal boron nitride (hBN) have attracted significant attention [8–12]. For example, a single layer of MoS$_2$ has high breaking strength (~23 GPa) and Young's modulus (~300 GPa), and hBN monolayer (like graphene) is characterized by excellent fracture strength (~70.0 GPa) and Young's modulus (~0.8 TPa) [13,14].

Because of these fascinating properties, 2D-nanosheet was widely applied to the preparation of high-performance polymer nanocomposites. Polar polymers have been extensively investigated, but polyolefin/MoS$_2$ and polyolefin/hBN nanocomposites have rarely been studied. Polyolefins interact unfavorably with the inorganic MoS$_2$ and hBN surfaces and incompatibility arises when these materials

are mixed with unmodified $MoS_2$ and hBN. The filler dispersion and interfacial interaction are crucial for improving the thermal and mechanical properties of polymer/inorganic hybrid nanocomposites.

Several researches have focused on surface modification of $MoS_2$ and hBN for polymer-based nanocomposites. Rajamathi et al. reported that amine-functionalized $MoS_2$ nanosheets showed effective dispersion in organic solvents. Interestingly, alkyl-modified $MoS_2$, such as octadecylamine (ODA) alkyl-chain groups typically exhibit strong interfacial adhesion with the polyolefin matrix [15]. The structure of hBN is similar to that of graphene and hence, hBN is used by modifying functional groups on hBN oxides [16,17].

Melt mixing represents a common method of preparing polymer-based nanocomposites, however, the dispersion of 2D nanosheets is not effective in this method. For in situ polymerization, the dispersion of 2D nanosheets was superior to that of melt mixing and solution mixing methods. The in situ polymerization approach, i.e., polymerization in the presence of nanofiller, shows significant promise for achieving improved nanofiller dispersion in a polymer matrix. A good dispersion is critical for the final properties of the nanocomposites [18,19].

Most in situ polymerizations of ethylene or propylene used the catalysts that reacted to the Mg compounds and $TiCl_4$ in turn on the surface of graphene oxide (GO). Ramazani et al. studied PE/graphene oxide (GO) nanocomposites, synthesized via an in situ polymerization method, with a Mg(OEt)2–GO-supported Ti-based Ziegler–Natta catalyst [20]. The effects of the degree of GO oxidation on the GO/BuMgCl-supported Ti-based Ziegler–Natta catalyst performance and PE/GO nanocomposite properties were also reported [21]. These catalysts are difficult to prepare and contain the desired amount of 2D nanosheets.

In this study, a novel method is used in preparation of supported catalysts containing 2D nanosheets without chemical reaction to the 2D-nanosheet's surface. Two-dimensional-nanosheet/ $MgCl_2$-supported Ti-based Ziegler–Natta catalysts are synthesized through a coagglomeration method for well-dispersed 2D-nanosheets in catalysts. The aggregation of individual 2D-nanosheet layers is prevented by applying the solid-state Ziegler–Natta catalyst during the preparation process. $MoS_2$ and hBN were modified with octadecylamine (ODA) and octyltriethoxysilane (OTES), respectively, for prevention of reaggregation and good compatibility with PP. Following the in situ polymerization of propylene, well-dispersed 2D-nanosheets PP composites were prepared. In addition, the effects of 2D-nanosheets on the catalyst performance and properties of PP nanocomposites were studied.

## 2. Results and Discussion

### 2.1. Characterization of Modified 2D-Nanosheets and Catalysts

The functionalized 2D-nanosheets, such as $MoS_2$-ODA and hBN-OTES, were confirmed via Fourier-transform infrared spectroscopy (FT-IR). As shown in Figure 1, several peaks corresponding to $MoS_2$-ODA were observed, but peaks associated with $MoS_2$ were absent. The spectra of $MoS_2$-ODA exhibited additional absorption peaks at 1020, 1110 and 1490 $cm^{-1}$ (aliphatic C-H bending) and the peaks at 2920 and 2840 $cm^{-1}$ (-$CH_2$- antisymmetric and symmetric stretching vibration). Characteristic peak at 1650 $cm^{-1}$ (N-H bending) shifted 1615 $cm^{-1}$ after surface modification of $MoS_2$ with ODA [22].

The broad B-N stretching was observed, including in-plane stretching vibration (B-N, 1378 $cm^{-1}$) and the out-of-plane bending mode (B-N, 809 $cm^{-1}$), as shown in Figure 2. After oxidation (hBN-OH), representative peaks such as those corresponding to hydroxyl group stretching (-OH, 3200–3500 $cm^{-1}$), in-plane stretching vibration (B-O, 1090 $cm^{-1}$) and the out-of-plane bending mode (B-O, 938 $cm^{-1}$) were also observed. Characteristic peaks of hBN-OTES occurred at 2850 $cm^{-1}$, 2925 $cm^{-1}$ (C-H stretching vibration), and 1100 $cm^{-1}$ (Si-O-C vibration), furthermore, the intensity of the –OH peak decreased significantly after surface modification of hBN-OH with OTES, indicating that hBN-OTES was successfully synthesized.

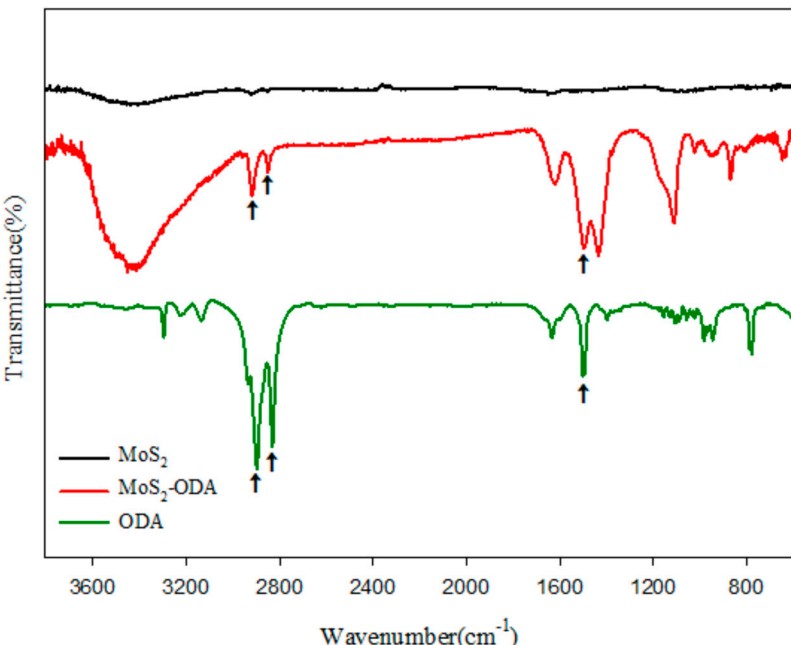

**Figure 1.** Fourier-transform infrared spectroscopy (FT-IR) spectra of $MoS_2$, $MoS_2$-ODA and octadecylamine (ODA).

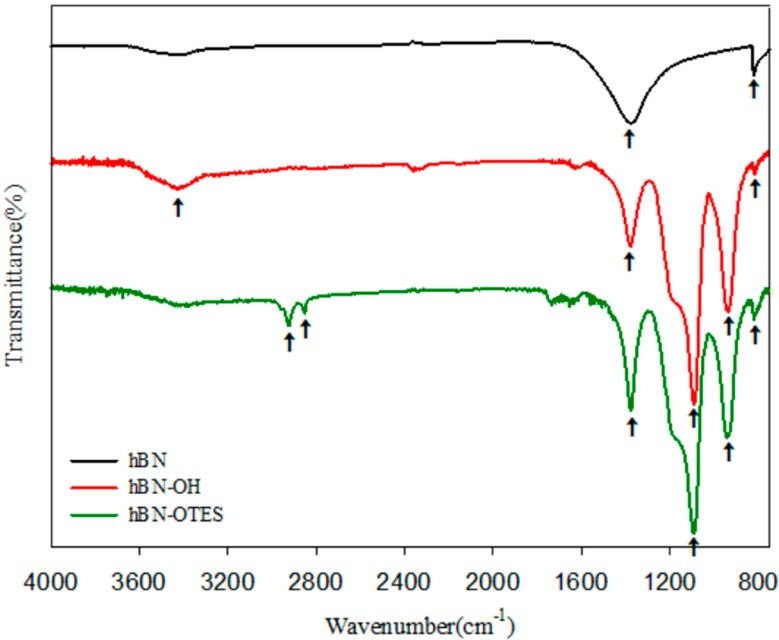

**Figure 2.** FT-IR spectra of hexagonal boron nitride (hBN), hBN-OH and hBN-OTES (octyltriethoxysilane).

The intercalation of $MoS_2$-ODA, hBN-OTES and the obtained catalysts were identified via X-ray diffraction (XRD).

An intense reflection peak at $2\theta = 14.3°$ (interlayer distance, i.e., *d*-spacing = 0.62 nm), which was observed for pristine $MoS_2$, was attributed to the (002) plane of $MoS_2$, as shown in Figure 3. After the treatment with n-BuLi, a new peak emerged at $2\theta = 7.8°$ (corresponding *d*-spacing: 1.13 nm). The diffraction peak of expanded $MoS_2$ occurred at 7.8°, while the peak of $MoS_2$-ODA (which formed after reaction between $MoS_2$ and ODA) occurred at 6.8° (corresponding *d*-spacing: ~1.3 nm). The peaks were weak and broad, indicating that most of the stacked $MoS_2$ layers could be efficiently intercalated

through reaction with ODA. The characteristic peak of ODA appeared at 19.5° and, it indicated the successful attachment of ODA to the $MoS_2$. For the $MoS_2$-ODA/ $MgCl_2$/ID/$TiCl_4$ catalyst, the (002) diffraction peak at 14.3° disappeared and only a broad peak could be observed. This confirmed that the $MoS_2$-ODA layers were surrounded by the clusters which formed $MgCl_2$ and $TiCl_4$ [17].

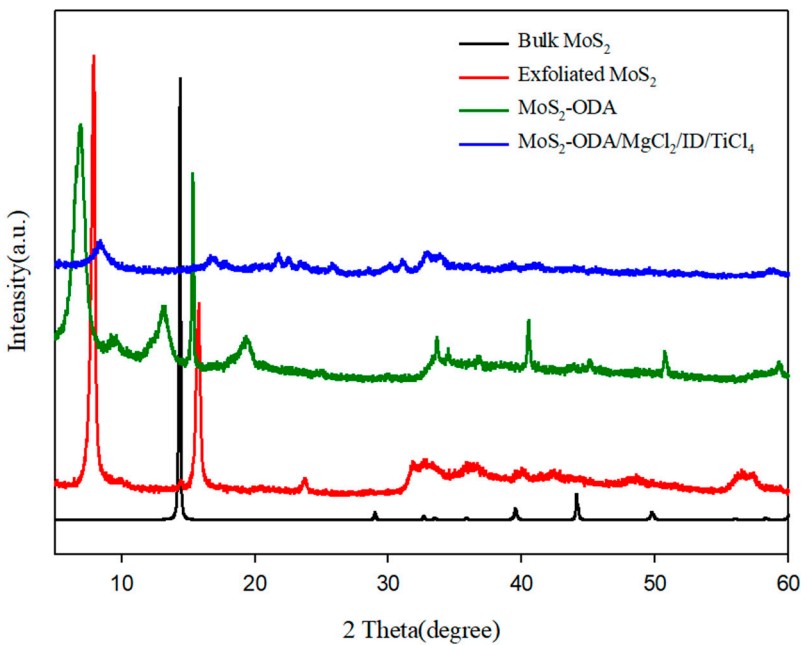

**Figure 3.** X-ray diffraction (XRD) pattern of bulk $MoS_2$, exfoliated $MoS_2$, $MoS_2$-ODA, and $MoS_2$-ODA/$MgCl_2$/ID/$TiCl_4$.

As shown in Figure 4, intense reflection was observed at $2\theta$ = 26.7°, 41.6°, 42.5°, 50.7°, 55.2° and 76.1° corresponding to the (002), (100), (101), (102), (004) and (110) planes of hBN skeleton. It indicates that the modification of hBN did not affect the crystalline structure of the hBN-OTES [23]. After oxidation of hBN, hBN-OH showed higher intensities for (002) and (004) peaks and a dramatically higher intensity ratio e.g., $I_{002}/I_{100}$ = 79.9 and $I_{004}/I_{100}$ = 3.5 as compared to hBN ($I_{002}/I_{100}$ = 23.7 and $I_{004}/I_{100}$ = 0.9), which might be attributed to the enhanced expore of the (002) basal plane because of the intercalation of few layered hBN-OH from hBN along the (002) plane [24,25]. The reflection peaks also corroborated the high intercalation of hBN-OTES, the intensity ratios were $I_{002}/I_{100}$ = 126.9 and $I_{004}/I_{100}$ = 4.4, which is a significant increase compared to hBN-OH, indicating the introduction of OTES on the surface of hBN-OH. In hBN-OTES/$MgCl_2$/ID/ $TiCl_4$, the (002) diffraction peak disappeared at 26.7° and only a broad peak was observed, indicating that the hBN-OTES layer is surrounded by clusters forming $MgCl_2$ and $TiCl_4$ [26].

The compositions of the obtained Ziegler–Natta catalysts in the absence and presence of 2D-nanosheets were confirmed by inductively coupled plasma (ICP). The Ti and Mg content of the $MgCl_2$/ID/$TiCl_4$ catalyst was 3.6 and 8.9 wt %, respectively. In the case of the $MoS_2$-ODA/$MgCl_2$/DIBP/$TiCl_4$ and hBN-OTES/$MgCl_2$/DIBP/$TiCl_4$ catalysts, the content of Ti were 3.7 and 3.6 wt % and the content of Mg were 8.7 and 8.6 wt %, which revealed similar [Ti]/[Mg] ratios for the catalysts with or without the 2D-nanosheets. The introduction of the nanosheets had negligible effect on the loading efficiency of $TiCl_4$ on the $MgCl_2$ support. Furthermore, coagglomeration methods yielded not chemical bonds between the 2D-nanosheets and $MgCl_2$, but resulted in $MgCl_2$ cluster coverage of the sheets.

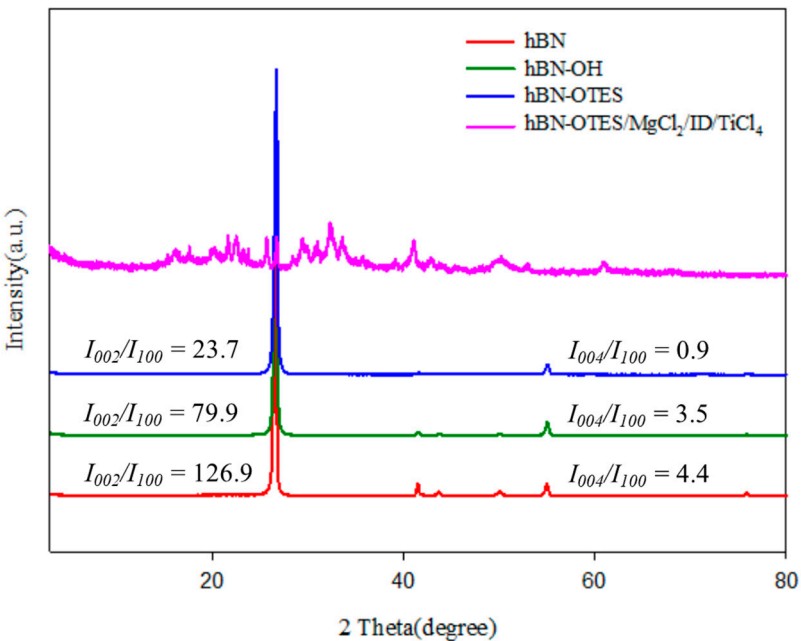

**Figure 4.** XRD pattern of hBN, hBN-OH, hBN-OTES and hBN-OTES/MgCl$_2$/ID/TiCl$_4$.

## 2.2. Preparation of PP/2D-Nanosheet Nanocomposites

To avoid aggregation of the 2D-nanosheets, the sheers were modified with long alkyl-compounds prior to preparation of the catalysts. The effects of the nanosheets on the catalyst performance and physical properties of PP were investigated. The propylene polymerization behaviors of the various 2D-nanosheets/MgCl$_2$/DIBP/TiCl$_4$ catalysts were evaluated with a triethylaluminum (TEA) cocatalyst.

As shown in Table 1, the activities of the MoS$_2$-ODA/MgCl$_2$/DIBP/TiCl$_4$ and hBN-OTES/MgCl$_2$/ DIBP/TiCl$_4$ catalysts were similar to that of a MgCl$_2$/DIBP/TiCl$_4$ catalyst under the same polymerization condition. As a result, the addition of 2D-nanosheets had no effect on the catalytic activity. Consider the reaction of MgCl$_2$ and 2D-nanosheets with TiCl$_4$ to form a dispersed solution. The assumption was that the formation of the Mg-Ti cluster covering the surface of the nanosheets preceded the reactions between the nanosheets and TiCl$_4$. Moreover, the Mg and Ti content and composition of the catalyst remained the same, with or without the incorporation of 2D-nanosheets into the catalysts.

**Table 1.** Polymerization results of 2D-nanosheet/MgCl$_2$-bisupported catalysts.

| Catalysts | Catalysts (mg) | [Al]/[Ti] | Time (min) | Isotactic Index (I.I) (%) | Activity (PP kg/g-Ti-h) | 2D-Nanosheets Contents (wt %) | Melt Index (M.I) |
|---|---|---|---|---|---|---|---|
| MgCl$_2$/DIBP/TiCl$_4$–TEA/DCPDMS | 200 | 5 | 60 | 93 | 0.68 | - | 0.53 |
| | 200 | 10 | 30 | 93 | 0.95 | | 0.62 |
| MoS$_2$-ODA/ MgCl$_2$/DIBP/TiCl$_4$–TEA/DCPDMS | 210 | 10 | 30 | 92 | 0.63 | 0.3 | 0.49 |
| | 250 | 10 | 60 | 93 | 0.90 | 0.9 | 0.45 |
| | 250 | 10 | 30 | 91 | 1.28 | 1.3 | 0.50 |
| | 400 | 30 | 30 | 92 | 0.96 | 7.0 | 0.52 |
| hBN-OTES/ MgCl$_2$/DIBP/TiCl$_4$–TEA/DCPDMS | 210 | 10 | 60 | 93 | 0.59 | 0.3 | 0.48 |
| | 250 | 10 | 60 | 94 | 1.13 | 0.7 | 0.45 |
| | 250 | 7 | 15 | 92 | 2.52 | 1.4 | 0.45 |
| | 400 | 10 | 45 | 91 | 1.12 | 4.0 | 0.49 |

Polymerization conditions: C$_3$H$_6$(1 atm), 40 °C, [ED]/[Al] = 0.2.

The isotactic index (I.I) values of PP nanocomposites containing 2D-nanosheets are similar to that of neat PP, which was obtained with the MgCl$_2$/DIBP/TiCl$_4$–TEA/DCPDMS catalyst system. The content of 2D-nanosheets in the nanocomposites was adjusted for different amounts (0.3–7.0 wt %) of catalysts and the polymerization time. The molecular weight was confirmed by the Melt Flow Index (M.I.) and M.I. values of 0.45 to 0.55 g/10 min were obtained.

## 2.3. Thermal Properties and Stabilities

The effect of 2D-nanosheets on the crystallization temperature ($T_c$), melting temperature ($T_m$) and degree of crystallinity ($X_c$) of PP composites (with different 2D-nanosheet content) was investigated via differential scanning calorimetry (DSC) (see Table 2 for the results).

**Table 2.** Thermal properties of Polypropylene/2-dimensional nanosheet (PP/2D-nanosheet) composites with various nanosheet contents.

| Catalysts | 2D-Nanosheets Contents (wt %) | $T_c$ (°C) | $T_m$ (°C) | $X_c$ (%) |
|---|---|---|---|---|
| $MgCl_2$/DIBP/$TiCl_4$ | - | 116.7 | 163.6 | 38.6 |
| $MoS_2$-ODA/ $MgCl_2$/DIBP/$TiCl_4$ | 0.3 | 119.0 | 164.6 | 43.4 |
| | 0.9 | 119.0 | 161.0 | 45.8 |
| | 1.3 | 122.0 | 163.0 | 55.5 |
| | 7.0 | 123.0 | 163.0 | 57.9 |
| hBN-OTES/ $MgCl_2$/DIBP/$TiCl_4$ | 0.3 | 123.8 | 164.0 | 45.4 |
| | 0.7 | 124.1 | 162.2 | 48.3 |
| | 1.4 | 124.9 | 163.8 | 53.6 |
| | 4.0 | 125.6 | 162.6 | 58.4 |

The $T_m$ of the PP obtained with $MgCl_2$/ID/$TiCl_4$ catalyst was 163 °C, and $T_m$ of the produced PP nanocomposites remained almost unchanged with incorporation of the 2D-nanosheets into the catalyst, although the $X_c$ increased gradually (from 38.6% to 58.4%) with increasing nanosheet content. The $T_c$ increased gradually (From 116.7 °C to 125.6 °C) with increasing 2D-nanosheet content of the nanocomposites, consistent with the result reported by Naffakh et al. [27]. They found that the $MoS_2$ had a considerable effect on the crystallization of the nanocomposite, and attributed this effect to the nucleation of $MoS_2$ on the monoclinic α–crystal form of PP. The result indicated that the crystallization temperature of PP has increased, owing to the nanosheets, which act as a nucleating agent.

The thermal stability of neat PP and PP/2D-nanosheets nanocomposites with different 2D-nanosheets was analyzed by thermogravimetric analysis (TGA) under nitrogen atmosphere (Figure 5 and Table 3). Compared with that of neat PP, the thermal stability of the obtained PP nanocomposites was significantly improved by the incorporation of 2D-nanosheets.

As shown in Figure 5, the TGA curves revealed a one-step degradation process and the thermal degradation temperatures increased with increasing 2D-nanosheet content. The degradation temperatures of PP/$MoS_2$-ODA nanocomposites ($T_{d5\%}$) are higher than that of neat PP. For 0.9, 1.3 and 7.0 wt % $MoS_2$-ODAcontent of PP/$MoS_2$-ODA nanocomposites, the $T_{d5\%}$ of composites was 14, 19 and 67 °C higher than that of PP. With regard to 1.3 wt % $MoS_2$-ODA loading, the maximum degradation temperature was 22 °C higher than that of PP. A distinct increase in thermal stability was due to $MoS_2$ being well-dispersed in the PP matrix. The introduction of 2D-nanosheets into PP can improve the thermal stability of PP, besides the fact that $MoS_2$ exhibits good thermal stability. The improvement in thermal stability can be attributed to the called "tortuous path" barrier effect of $MoS_2$, which delays the escape of volatile degradation products and thus slows down the initial degradation step [28].

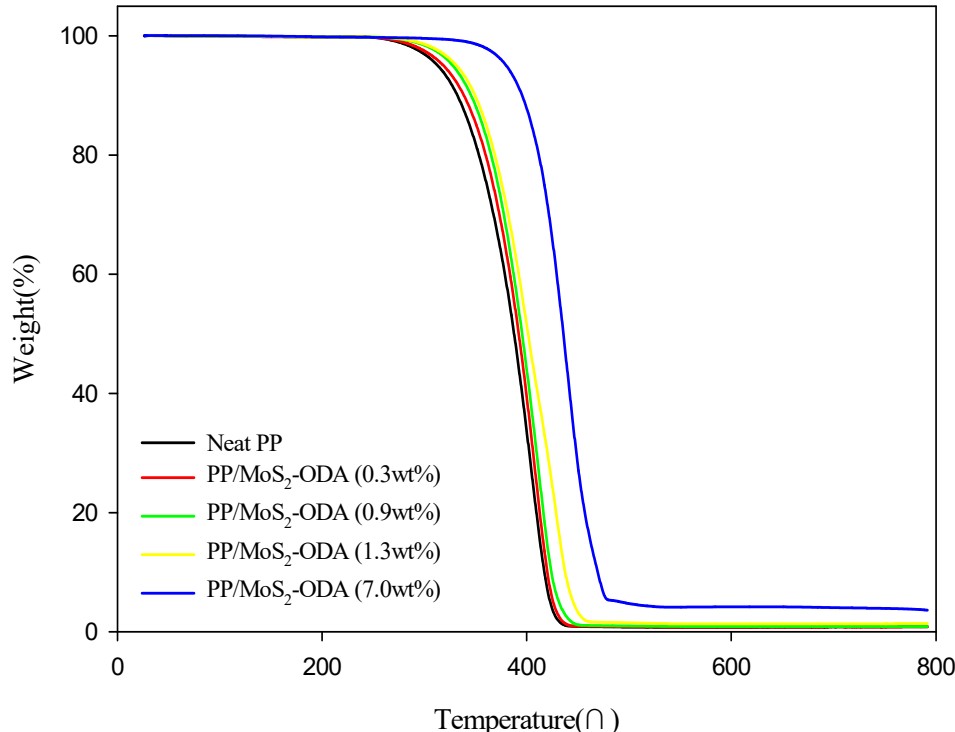

**Figure 5.** Thermogravimetric analysis (TGA) curves of neat PP and PP/MoS$_2$-ODA nanocomposites with various MoS$_2$-ODA contents.

**Table 3.** Thermal stability of PP/MoS$_2$-ODA composites at various contents of 2D-nanosheet.

| Samples | $T_{d5\%}$ (°C) | $T_{d10\%}$ (°C) | $T_{dmax}$ (°C) | Char Yield (wt %) |
|---|---|---|---|---|
| Neat PP | 313 | 332 | 404 | 0.8 |
| PP/MoS$_2$-ODA (0.3 wt %) | 318 | 338 | 409 | 0.8 |
| PP/MoS$_2$-ODA (0.9 wt %) | 327 | 345 | 412 | 0.9 |
| PP/MoS$_2$-ODA (1.3 wt %) | 331 | 349 | 426 | 1.4 |
| PP/MoS$_2$-ODA (7.0 wt %) | 380 | 395 | 441 | 3.6 |
| PP/hBN-OTES (0.3 wt %) | 325 | 345 | 416 | 0.6 |
| PP/hBN-OTES (0.7 wt %) | 331 | 349 | 414 | 1.0 |
| PP/hBN-OTES (1.4 wt %) | 319 | 338 | 411 | 1.4 |
| PP/hBN-OTES (4.0 wt %) | 325 | 346 | 419 | 3.2 |

*2.4. Mechanical Properties*

The mechanical properties of the PP/2D-nanosheet composites with various 2D-nanosheet content are presented in Figure 6 and Table 4.

With the incorporation of 2D-nanosheets into the nanocomposites, the tensile strength and modulus increased by 40% and 80% relative to those of neat PP. Even with just a small amount (0.3 wt %) of these nanosheets in the PP nanocomposites, the tensile strength and modulus increased considerably without a significant change in the elongation at break. Two-dimensional–nanosheet loadings of 0.3 wt % and 1.3–1.4 wt % yielded tensile strength increases of 18% and 34% as well as modulus increases of 13% and 58%, respectively, relative to the values of the neat PP. These results indicate the remarkable stiffness-toughness balance of PP/2D-nanosheet composites produced via the in situ polymerization method with 2D-nanosheets/MgCl$_2$/ID/TiCl$_4$ catalysts prepared using a coagglomeration method. This result suggested that 2D-nanosheets exert a strong reinforcement effect in the composites, owing to the well-dispersed 2D-nanosheets and good compatibility between 2D-nanosheet and PP. The improvement of mechanical properties was due to the 2D-nanosheets absorption of fracture energy and prevention of crack propagation by the 2D-nanosheets [29].

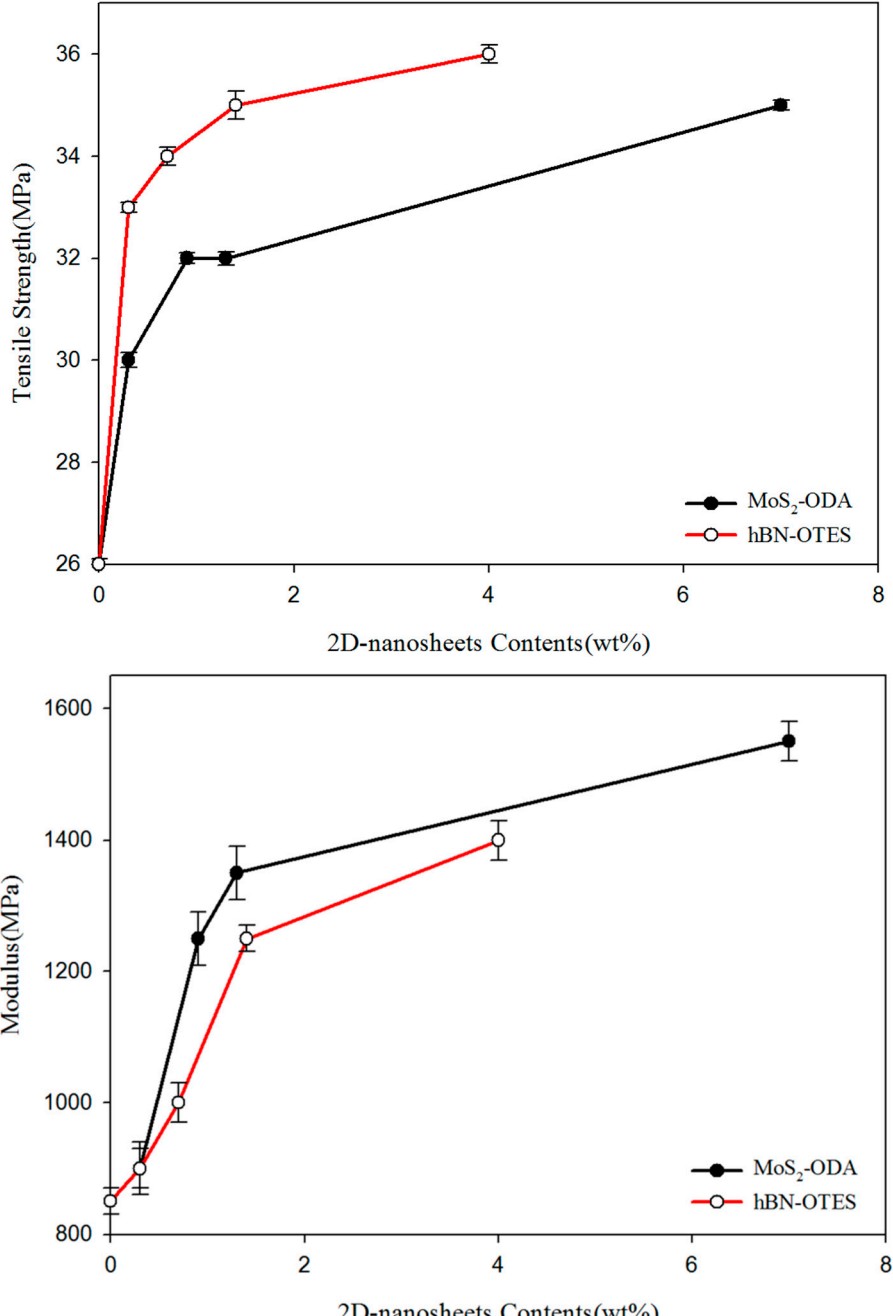

**Figure 6.** Effect of 2D-nanosheet contents on the tensile strength and Young's modulus of PP/2D-nanosheets.

Generally, the dispersion and interfacial adhesion between 2D-nanosheets and polymer matrix occupy critical roles in enhancing the properties of nanocomposites. To evaluate the dispersion of 2D-nanosheets in PP matrix, scanning electron microscopy (SEM) was employed to observe the fractured surface of nanocomposites. For neat PP, the cross section is smooth with noncrinkled morphology, whereas the fractured surfaces of the PP/2D-nanosheet nanocomposite became comparative roughness in Figure 7. Two-dimensional-nanosheets are dispersed uniformly without obvious reaggregation. The morphological difference reveals that the modification of ODA and OTES is beneficial to promoting the dispersity of 2D-nanosheets and improving the interaction between 2D-nanosheets and the PP matrix.

**Table 4.** Mechanical properties of the PP/2D-nanosheet nanocomposites at various contents of 2D-nanosheet.

| Samples | Tensile Strength (MPa) | Modulus (MPa) | Elogation at Break |
|---|---|---|---|
| Neat PP | 26.0 ± 1 | 850 ± 20 | ≥500 |
| PP/MoS$_2$-ODA (0.3 wt %) | 30.0 ± 1 | 900 ± 30 | ≥500 |
| PP/MoS$_2$-ODA (0.9 wt %) | 32.0 ± 1 | 1250 ± 40 | ≥500 |
| PP/MoS$_2$-ODA (1.3 wt %) | 32.0 ± 1 | 1350 ± 40 | ≥500 |
| PP/MoS$_2$-ODA (7.0 wt %) | 35.0 ± 1 | 1550 ± 30 | ≥500 |
| PP/hBN-OTES (0.3 wt %) | 33.0 ± 1 | 900 ± 40 | ≥500 |
| PP/hBN-OTES (0.7 wt %) | 34.0 ± 1 | 1000 ± 30 | ≥500 |
| PP/hBN-OTES (1.4 wt %) | 35.0 ± 1 | 1250 ± 20 | ≥500 |
| PP/hBN-OTES (4.0 wt %) | 36.0 ± 1 | 1400 ± 30 | ≥500 |

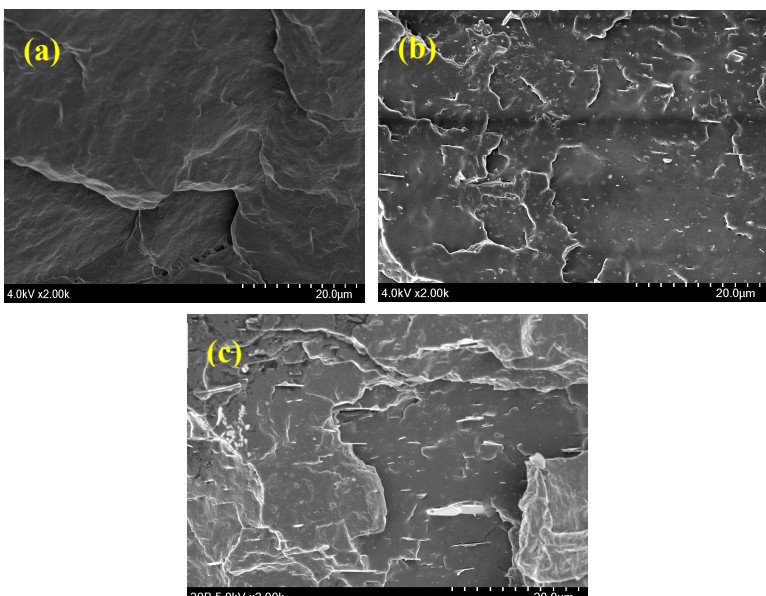

**Figure 7.** SEM images of the fractured surfaces of (**a**) neat PP (**b**) PP/MoS$_2$-ODA (7.0 wt %) and (**c**) PP/hBN-OTES (4.0 wt %).

## 3. Experimental Section

### 3.1. Materials

Molybdenum disulfide (MoS$_2$, Sigma Aldrich, Saint Louis, MO, USA), hexagonal boron nitride (hBN, Alfa Aesar, Ward Hill, MA, USA), n-butyllithium, octadecylamine (ODA, TCI, Tokyo, Japan), octyltriethoxysilane (OTES, Sigma Aldrich, Saint Louis, MO, USA), anhydrous magnesium chloride (MgCl$_2$, Sigma Aldrich, Saint Louis, MO, USA), 2-ethyl-1-hexanol (EHA, Sigma Aldrich, Saint Louis, MO, USA), diisobutylphthalate (DIBP, Sigma Aldrich, Saint Louis, MO, USA) triethylaluminum (TEA, Tosho Akzo, Japan), titanium tetrachloride (TiCl$_4$, Daesung Chemical & Metal Co., Gyeonggi, Korea), and dicyclopentyldimethoxysilane (DCPDMS, Korea Petrochemical Ind. Co., Korea) were used as received. n-Hexane (n-Hx, Duksan Chemical Co., Korea), dimethyl formamide (DMF, Duksan Chemical Co., Korea) and tetrahydrofuran (THF, Duksan Chemical Co., Korea) were purified with sodium/benzophenone under nitrogen prior to use.

### 3.2. Synthesis of MoS$_2$-ODA and hBN-OTES

MoS$_2$ was modified in accordance with a previously reported method [9,21]. Scheme 1 shows the procedure for MoS$_2$-ODA. For this preparation, MoS$_2$ (2 g) was placed in an autoclave and

n-butyllithium (15 mL, 2.5 M in n-hexane) was added. The autoclave was heated at 150 °C for 12 h under a $N_2$ atmosphere. The autoclave was cooled to room temperature and the product was filtered and washed with n-hexane (10 × 50 mL). Afterward, the obtained $MoS_2$ immersed in ODA solution (10 g in 100 mL EtOH) under 2 h of ultra-sonication at ambient temperature, thereby producing a suspension of ODA-modified $MoS_2$ ($MoS_2$-ODA). The suspension was neutralized with HCl and the products were washed with hot EtOH to remove unreacted ODA. The $MoS_2$-ODA powder was obtained by freeze-drying. (Scheme 1)

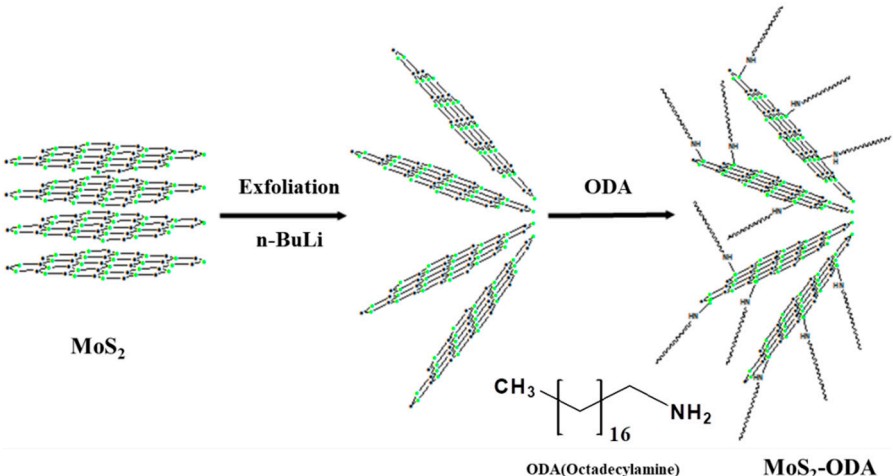

**Scheme 1.** Synthesis of $MoS_2$-ODA.

Few-layered hBN was prepared from the bulk hBN via a liquid-assisted mechanical exfoliation method as follows; [30,31] 2 g of the bulk hBN powder was added to a solution of 100 mL isopropanol and 100 mL deionized water. The mixture was mechanically stirred and simultaneously sonicated using ultrasonic bath at 80 W for 10 h. The dispersions were centrifused to remove stacked hBN powders. The few layered hBN in the resultant supernatant was collected by centrifugation and then dried at 60 °C for 24 h. The mass of few-layered hBN was approximately 0.25 g.

hBN oxide (hBN-OH) was prepared from hBN following a modified Hummers' oxidation method. hBN-OH was prepared as follows: hBN (0.5 g) and $KMnO_4$ (6 g) were mixed and a mixture of phosphoric acid/sulfuric acid (1/8 wt/wt, 150 mL) was added to this mixture. The oxidation was performed at 140 °C for 24 h under reflux, and the product was washed with $H_2O$/EtOH and HCl solution. The synthetic procedure of alkylsilane modified hBN (hBN-OTES) is illustrated in Scheme 2. During the process, hBN-OH (0.1 g) was dispersed in EtOH/$H_2O$ (95/5 vol/vol, 100 mL) via 2 h of ultra-sonication and OTES (3 mL) was subsequently added to the dispersion. The mixture was stirred for 24 h at 90 °C under reflux and EtOH was then added to remove unreacted silane compounds. Afterward, the obtained products were dried under vacuum at 100 °C. (Scheme 2)

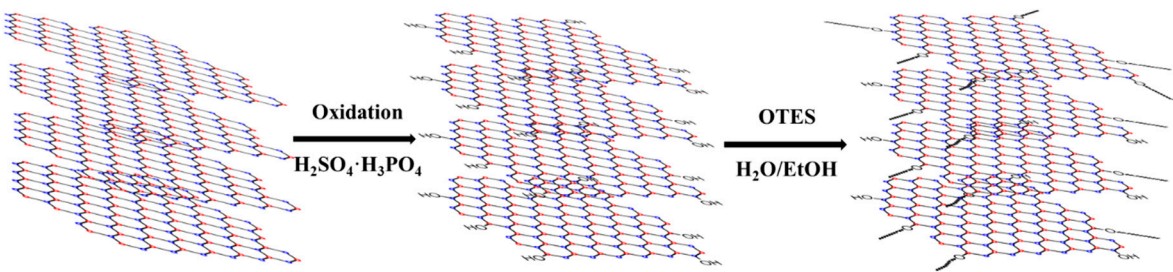

**Scheme 2.** Synthesis of hBN-OTES.

### 3.3. Synthesis of 2D-Nanosheets/MgCl$_2$-Bisupported Ziegler–Natta Catalyst

2D-nanosheet/MgCl$_2$-bisupported Ziegler–Natta catalysts were prepared via a coagglomeration method [17]. During the process, MgCl$_2$ (1 g) and the nanosheets were added to EHA (19.6 mL) and DIBP (0.42 mL, as an internal donor) under N$_2$. The reaction mixture was heated to 160 °C to obtain a well-dispersed 2D-nanosheet suspension and the obtained suspension was ultrasonicated for 5 h. Subsequently, TiCl$_4$ (23 mL) was dropped to the 2D-nanosheets dispersed MgCl$_2$ solution; precipitation was observed during this process. After 2 h, the precipitate was filtered to remove unreacted TiCl$_4$ and then washed repeatedly with n-hexane and re-dispersed in the n-hexane.

### 3.4. Preparation of the PP/2D-Nanosheet Nanocomposites via In-Situ Polymerization

Polymerization was carried out in a 300 mL glass reactor, the reactor was replaced with N$_2$ and charged with the required amount of n-hexane, which was stirred at 40 °C under propylene (1 atm) for the desired period of time. Afterward, the cocatalyst (TEA) and DCPDMS (external donor) were added. The polymerization was initiated by injection catalyst under a continuous propylene flow. A bubbler was used to maintain a constant propylene pressure during the polymerization. After the desired time, the product was poured into methanol, then filtered and dried at 80 °C in vacuo.

### 3.5. Characterizations

The chemical structures of the 2D-nanosheets were characterized via the transparent mode of Fourier transform infrared (FT-IR) spectroscopy (Jasco 4100, Tokyo, Japan). The spectra of samples were performed in the wavenumber range of 4000–500 cm$^{-1}$ at 128 scans per and 4 cm$^{-1}$ resolution using KBr pellet method. X-ray diffraction (XRD) patterns of the nanosheets and catalysts were obtained on Phillips X-Pert PRO MRD diffractometer (Cu-K$\alpha$ radiation). In the case of catalysts, the amorphous cover glass was covered to avoid oxygen and the 2θ scanning range was 5~80° with a rate of 4°/min. The Mg and Ti contents of the 2D-nanosheet/MgCl$_2$-bisupported Ziegler–Natta catalysts were determined through inductively coupled plasma atomic emission spectroscopy (ICP-AES, PerkinElmer, Optima 7300DV) The HNO$_3$:HCL:HF (5:15:3 (v/v)) solution was used for the pretreatment of catalysts.

The polymer product was fractionated via a 12 h extraction with boiling n-heptane, to determine the corresponding isotactic index (I.I.), i.e., the weight percentage of n-heptane-insoluble polymer excluding the content of 2D-nanosheet. The 2D-nanosheets content of the nanocomposite was calculated by the weight of the obtained PP and the catalyst injected during polymerization. The melting temperature ($T_m$) crystallization temperature ($T_c$) and degree of crystallinity ($X_c$) of the obtained polymer and nanocomposites were determined by means of differential scanning calorimetry (DSC, Setaram, Caluire, DSC131evo, France) measurements (heating and cooling rate; 10 °C/min). Furthermore, the decomposition temperature was determined under N$_2$ atmosphere with a Setaram Labsys evo thermogravimetric analysis (TGA), employing a programmed heating rate of 10 °C/min from 30 °C to 800 °C [32–35].

The melt index of the obtained polymers was measured using a melt flow indexer (MFI 10, Davenport Co.) at 190 °C (ASTM D1238).

The mechanical properties of 4.0 × 50.0 × 1.0 mm$^3$ (length × width × thickness) PP and PP/2D-nanosheet nanocomposites specimens were measured (cross-head speed: 30.0 mm/min) with a universal testing machine (SALT, ST-1001).

## 4. Conclusions

2D-nanosheet/MgCl$_2$/ID/TiCl$_4$ catalysts were prepared through a coagglomeration method for achieving well dispersed 2D-nanosheets in a catalyst. PP/2D-nanosheet nanocomposites were fabricated via in situ polymerization using these catalysts. Moreover, MoS$_2$ and hBN surfaces were modified with alkylamine (octadecylamine, ODA) and octyltriethoxysilane (OTES) for good dispersion in the catalysts.

Two-dimensional–nanosheets were incorporated into the catalyst, and a Ti/Mg molar ratio of 0.4–0.5 was realized with or without incorporation of 2D-nanosheets. Furthermore, the incorporation of these nanosheets had no effect on the loading efficiency of $TiCl_4$ and formation of $MgCl_2$-$TiCl_4$ clusters during preparation of the catalyst.

The $T_m$ of the produced PP nanocomposites was almost unchanged even after incorporation of 2D-nanosheets into the catalyst. However, the $T_c$ and $X_c$ increased gradually with increasing 2D-nanosheet content. The nanosheets had a significant effect on the crystallization of the nanocomposite, owing to the nucleating effect on the monoclinic $\alpha$–crystal form of PP. The significant increase in thermal stability was due to $MoS_2$-ODA well-dispersed in the PP matrix. Moreover, the MoS2 are characterized by a high heat capacity and can form heat-resistant layers that act as mass transport barriers, which will delay the onset of volatile degradation.

The mechanical properties were improved via the incorporation of 2D-nanosheets. For example, the tensile strength and Young's modulus of the composites with 0.3 wt % $MoS_2$-ODA were 40% and 80% higher, respectively, than those of neat PP. This indicates that the dispersion of 2D-nanosheets and compatibility with PP are excellent. The preparation of Ziegler–Natta catalysts via the coagglomeration method allows easy introduction and content adjustment of various 2D-nanosheets. This method represents a straightforward means of preparing PP nanocomposites with high contents of 2D-nanosheets for high-performance PP.

**Author Contributions:** Conceptualization, H.-x.Z.; K.-B.Y.; Investigation, B.-G.S.; K.-B.Y.; Validation, D.-E.L.; H.-x.Z.; Writing—original draft preparation, D.-E.L.; K.-B.Y.; Writing—review and editing, H.-x.Z.; B.-G.S.; D.-E.L.; K.-B.Y.; All authors have read and agreed to the published version of the manuscript.

**Funding:** This work was supported by the National Research Foundation of Korea (NRF) grant funded by the Korea government (MIST) (No. NRF-2018R1A5A1025137 and No. NRF-2019R1A2C3003890).

**Conflicts of Interest:** The authors declare no conflict of interest.

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
