# Peer review of "Preparation of PP/2D-Nanosheet Composites Using MoS2/MgCl2- and BN/MgCl2-Bisupported Ziegler–Natta Catalysts"

_catalysts, doi:10.3390/catal10060596_

Round 1

Reviewer 1 Report

The authors prepared PP-based nanocomposites with 2D nanosheets (MoS2 and hBN) by using 2D nanosheet/MgCl2 bi-supported Ziegler-Natta catalyst system in an in-situ manner. The catalysts were synthesized by co-aggregation method with alkyl modified 2D nanosheets, and the activity was not damaged by the presence of the nanofillers. The obtained PP nanocomposites exhibited a better mechanical property compared to the pristine PP.

I recommend that this paper be rejected. I believe there should be a more appropriate journal for this manuscript.

Major comments:

#1. The scope of this journal is of catalysts and catalyzed reactions. This manuscript however seems to focus on the property of the obtained PP nanocomposites since the characterization of catalysts as well as investigation on the catalyst performance are, I am afraid to say, not so deeply done, in which the readers of this journal would be less interested.  

#2. The motivation to use 2D nanosheets was not specifically explained in the introduction. It is not clear what advantages 2D nano fillers have compared to the conventional spherical nano fillers.

#3. The novelty of in-situ preparation is not clear. 2D nanosheets/PP nanocomposites prepared in an ex-situ manner would make the discussion more concrete.

#4. The authors stress that the improvement of physical properties is due to the good dispersion of the 2D nanosheets. However, no evidence supports this conclusion. TEM observation and/or rheological analysis approach are indispensable for discussion regarding dispersion.

#5. I could not find good reasons to perform thermal degradation test by TG analysis under inert atmosphere. PP is a commodity plastic, which is used in the living environment, therefore thermal and/or light induced oxidative degradation is important in terms of lifetime of the material. Improving combustion resistance is another important topic. In any case, the presence of oxygen is essential. Moreover, the decomposition temperature that authors reported is over 300 ºC, where PP should be completely molten. If the matrix were heat resistance resins such as polyimides, it would be worth to do because such engineering resins may be used in an extraordinary environment. I would recommend the authors to give justifications for TG analysis.

Minor comments:

#6. The methods for FT-IR spectra of the powder samples should be explained in 3.5. Characterization. I believe it must be measured via DRIFT or ATR-IR etc.

#7. In Figure 7, MoS2-ODA has several additional peaks. Especially, the intense peak appeared next to the N-H stretching vibration peak (1490 cm-1) is not discussed. Besides, the peaks at finger-print region (ca 1500-500 cm-1) are totally different from ODA. I recommend the authors to investigate the spectra more carefully.

#8. MgCl2 supported Ziegler-Natta catalysts are air/moisture sensitive. How to prevent moisture contamination during XRD measurement should be described.

#9. 2D nanosheet content in the bi-supported catalysts should be determined. The authors performed ICP analysis on the catalysts, so it should be readily calculated. Pretreatment for ICP analysis should also be described in 3.5. Characterization.

#10. (002) diffraction peak was almost unchanged even after OTES treatments, which means exfoliation was not occurred at least at this step. Therefore, scheme 2 is quite misleading.

#11. Regarding 3.1. Materials, it is necessary to state where the materials obtained from.

#12. In scheme 2, the image of hBN seems to be overlapped with the other figure. This should be corrected.

#13. In-situ prepared PP nanocomposites include the nanofiller. How to subtract the filler content from I.I. calculation should be explained.

Author Response

Thanks for your detailed review of this manuscript. According the reviewer’s comments, I responded in turn with a point-by-point.                                                         

Comment 1.                        

Although this article represented the properties of PP nanocomposite, which composite was prepared with a novel bisupporter catalyst via in situ polymerization of propylene. The bisupported catalysts prepared by co-agglomeration method. This co-agglomeration method is different from the conventional catalyst manufacturing, and can effectively introduce 2D-nanosheets.

This is explained in addition to Introduction part as below. (Revised manuscript, Line 53-61)

Most in situ polymerization of ethylene or propylene used the catalysts that reacted to the Mg compounds and TiCl4 in turn on the surface of graphene oxide (GO). Ramazani et al. studied PE/graphene oxide (GO) nanocomposites, synthesized via an in situ polymerization method, with a Mg(OEt)2–GO-supported Ti-based Ziegler–Natta catalyst [16]. The effects of the degree of GO oxidation on the GO/BuMgCl-supported Ti-based Ziegler–Natta catalyst performance and PE/GO nanocomposite properties were also reported [17]. These catalysts are difficult to preparation and contain the desired amount of 2D nanosheets.

In this study, a novel method is used in preparation of supported catalysts containing 2D nanosheets without chemical reaction to 2D nanosheet’s surface. 2D-nanosheet/MgCl2-supported Ti-based Ziegler–Natta catalysts are synthesized through a co-agglomeration method for well-dispersed.  

Comment 2.

Many studies have reported that 2D nanosheet is more effective than spherical nanomaterials in improving the properties of the nanocomposites, so it has not been explained in Introduction. If the reviewer is necessary, we will add the spherical nanomaterials to the Introduction part.

Comment 3.

The in situ polymerization method does not have a novelty, the composition of the catalyst used to obtain PP nanocomposites by co-agglomeration method has novelty. The co-agglomeration of bi-supported catalyst containing 2D-nanosheet is unique.

Inserted in Revised manuscript, Line 53-61

Comment 4.

The dispersion of 2D nanosheet is identified by the SEM and shown in Figure 7. The results are described as below:

Generally, the dispersion and interfacial adhesion between 2D-nanosheets and polymer matrix occupy critical roles in enhancing the properties of nanocomposites. To evaluate the dispersion of 2D-nanosheets in PP matrix, SEM was employed to observe the fractured surface of nanocomposites. For neat PP, the cross-section is smooth with no-crinkled morphology, whereas the fractured surfaces of the PP/2D-nanosheet nanocomposite became comparative roughness in Fig. 7. 2D-nanosheets are dispersed uniformly without obvious aggregation. The morphological difference reveals that the modification of ODA and OTES is beneficial to promoting the dispersity of 2D-nanosheets and improving the interaction between 2D-nanosheets and PP matrix.

Inserted in Revised manuscript, Line 199-210

Comment 5.

TG analysis of composites is a common method to measure under the conditions of inert atmosphere. Melt processing of plastic or composites is also performed under the inert atmosphere. When examining the degree of oxidation, it is common to perform TG anlaysis under oxygen conditions. If necessary, I will measure and present under oxygen conditions.

Comment 6.

The measurement method is corrected as follow and described in 3.5. Characterization. The chemical structures of the 2D-nanosheets powder were characterized via the transparent mode of Fourier transform infrared (FT-IR) spectroscopy.

Inserted in Revised manuscript, Line 262

Comment 7.

The FT-IR spectra analysis is corrected as follows and described in 2.1. Characterization of modified 2D-nanosheets and catalysts.

As shown in Figure 1, several peaks corresponding to MoS2-ODA were observed, but peaks associated with MoS2 were absent. The spectra of MoS2-ODA exhibited additional absorption peaks at 1020, 1110 and 1490 cm-1 (aliphatic C-H bending) and the peaks at 2920 and 2840 cm-1 (-CH2- antisymmetric and symmetric stretching vibration). Characteristic peak at 1650 cm-1 (N-H bending) shifted 1615 cm-1 after surface modification of MoS2 with ODA

Inserted in Revised manuscript, Line 73-76

Comment 8.

This part is corrected and explained in 3.5. as follows; X-ray diffraction (XRD) patterns of the nanosheets and catalysts powder were obtained on Phillips X-Pert PRO MRD diffractometer (Cu-Kα radiation) in a nitrogen atmosphere.

Inserted in Revised manuscript, Line 264-265

Comment 9.

The ICP results are calculated and described in 2.1. Characterization of modified 2D-nanosheets and catalysts as follows:

The compositions of the obtained Ziegler–Natta catalysts in the absence and presence of 2D-nanosheets were confirmed by ICP. The Ti and Mg content of the MgCl2/ID/TiCl4 catalyst was 3.6 and 8.9 wt%, respectively. In case of MoS2-ODA/MgCl2/DIBP/TiCl4 and hBN-OTES/MgCl2/DIBP/TiCl4 catalysts, the content of Ti were 3.7 and 3.6 wt% and the content of Mg were 8.7 and 8.6 wt%, which revealed similar [Ti]/[Mg] ratios for the catalysts with of without the 2D-nanosheets.  

Inserted in Revised manuscript, Line 117-120

Pretreatment process for ICP analysis is described in 3.5 as follows: The HNO3:HCL;HF (5:15:3 (v/v)) solution was used for the pre-treatment of catalysts.

Inserted in Revised manuscript, Line 267-268

Comment 10.

When evaluating the intercalation or exfoliation of hBN by XRD, it is not confirmed by shift of peak but by change of intensity of peak. An explanation of the intensity is given in 2.1. Characterization of Modified 2D-nanosheets and Catalysts as follows:

As shown in Figure 4, the reflection patterns of hBN. hBN-OH and hBN-OTES were observed at 2θ = 26.7°, 41.6°, 42.5°, 50.7°, 55.2° and 76.1° corresponding to the (002), (100), (101), (102), (004) and (110) planes, respectively. After oxidation of hBN, hBN-OH showed higher intensities for (002) and (004) peaks and a dramatically higher intensity ratio e.g. I002/I100 = 79.9 and I004/I100 = 3.5 as compared to hBN (I002/I100 = 23.7 and I004/I100 = 0.9), which might be attributed to the enhanced expore of the (002) basal plane because of the intercalation of few layered hBN-OH from hBN along the (002) plane. The reflection peaks also corroborated the high intercalation of hBN-OTES, the intensity ratios were I002/I100 = 126.9 and I004/I100 = 4.4, which is a significant increase compared to hBN-OH, indicating the introduction of OTES on the surface of hBN-OH.

Inserted in Revised manuscript, Line 103-111

Comment 11.

The company that purchased the materials is indicated in 3.1. Materials part.

Inserted in Revised manuscript, Line 214-221

Comment 12.

Scheme 2 is redrawn and represented.

Inserted in Revised manuscript, Line 114-115

Comment 13.

It is revised according to the comment and described in 3.5. as follows.

The polymer product was fractionated via a 12 h extraction with boiling n-heptane, to determine the corresponding isotactic index (I.I.), i.e., the weight percentage of n-heptane-insoluble polymer excluding the content of 2D-nanosheet. The 2D-nanosheets content of the nanocomposite was calculated by the weight of the obtained PP and the catalyst injected during polymerization.

Inserted in Revised manuscript, Line 270-272

Reviewer 2 Report

In this manuscript the authors reported about fabrication of  a PP/MoS2 and PP/hBN nanocomposites with varying concentration via in-situ polymerization using 2D-nanosheet/MgCl2-supported Ti-based Ziegler–Natta catalysts. The subject of this paper is interesting and promising since the properties render 2D-nanomaterials an attractive substitute for the fabrication of high-performance polymer nanocomposites.  

More in depth, 2D-nanosheet/MgCl2-supported Ti-based Ziegler–Natta catalysts are synthesized. The aggregation of individual 2D-nanosheet layers is prevented by applying the solid-state Ziegler-Natta catalyst during the preparation process. Following the in-situ polymerization of propylene, well-dispersed 2D-nanosheets PP composites were prepared. In addition, the effects of 2D-nanosheets on the catalyst performance and properties of PP nanocomposites were studied.

So the main goal of the manuscript is to provide a  facile way of preparing PP nanocomposites with high contents of 2D-nanosheets for high-performance PP. Moreover the authors reported that the mechanical properties of produced PP were improved by the incorporation of 2D-nanosheets.   

The manuscript is well written (just few typos here and there, that can be easily fixed) and the results are well commented and supported. I suggest the publication of the manuscript as it is.

Author Response

Reviewer 2 did not have any particular comments, but thank you for your detailed review of this manuscript.

Reviewer 3 Report

This review concerns the article Preparation of PP/2D-nanosheet composites using MoS2/MgCl2- and

BN/MgCl2-bisupported Ziegler-Natta catalysts (Manuscript ID: catalysts-787133).

The work concerns synthesis and characteristic of polypropylene composites with molybdenum disulfide and boron nitride. The nanocomposites were prepared through an in-situ polymerization using MgCl2-supported Ti-based Ziegler-Natta catalysts. Molybdenum disulfide and boron nitride were modified with octadecylamine and octyltriethoxysilane and modification were confirmed by FT-IR spectra. The intercalation of MoS2-ODA, hBN-OTES and the obtained catalysts were identified via XRD. Polymerization results of such supported catalysts. Thermal properties and stabilities of obtained materials were analyzed mainly using DSC technique. The in-situ polymerization approach is interesting and worth to be presented to scientific community.

The work is written clearly and concise. In my opinion the manuscript should be accepted for publication.

Minor remarks:

  1. The activities of the MoS2-ODA / MgCl2 / DIBP / TiCl4 and hBN-OTES / MgCl2 / DIBP / TiCl4 and MgCl2 / DIBP / TiCl4 catalysts were tested at a different Al / Ti ratio and the amount of typical Mg-Ti catalyst. Could this have a significant impact on the final productivity of the catalysts?
  2. The work would be enriched with SEM / EDS photos of sample composites showing the distribution of nanoparticles in the composite.

Author Response

Thanks for your detailed review of this manuscript.

Comment 1.

The data of catalyst activity under the same polymerization conditions were inserted in Table 1.

Inserted in Revised manuscript, Line 138

Comment 2.

The dispersion of 2D nanosheet is identified by the SEM and shown in Figure 7.

Generally, the dispersion and interfacial adhesion between 2D-nanosheets and polymer matrix occupy critical roles in enhancing the properties of nanocomposites. To evaluate the dispersion of 2D-nanosheets in PP matrix, SEM was employed to observe the fractured surface of nanocomposites. For neat PP, the cross-section is smooth with no-crinkled morphology, whereas the fractured surfaces of the PP/2D-nanosheet nanocomposite became comparative roughness in Fig. 7. 2D-nanosheets are dispersed uniformly without obvious aggregation. The morphological difference reveals that the modification of ODA and OTES is beneficial to promoting the dispersity of 2D-nanosheets and improving the interaction between 2D-nanosheets and PP matrix.

Inserted in Revised manuscript, Line 199-210

Round 2

Reviewer 1 Report

I appreciate your sincere responses to my comments and relevant efforts. I changed my recommendation to “reconsider after major revision” from “rejection”. Regarding your answers for comments #1, #3, #4, #7, #9, #11, #12, #13, I have no further comments. On the other hand, the other answers still have spaces for improvement. In particular, I have a relatively larger concern on the authors’ answer for comment #10 even though it started from a minor comment.

Comment 2.

Authors’ answer:

Many studies have reported that 2D nanosheet is more effective than spherical nanomaterials in improving the properties of the nanocomposites, so it has not been explained in Introduction. If the reviewer is necessary, we will add the spherical nanomaterials to the Introduction part.

Maybe my comment was not clear and made the authors misunderstood. I did not recommend to increase the number of citations regarding spherical fillers. Rather, I recommended specifying the advantages of 2D nanofillers. The keywords would be gas barrier property, aspect ratio, interaction among fillers, permittivity, and etc. I believe the stiffness or strength of the filler itself does not matter in polyolefine based composites because most of the inorganic fillers are far stronger than the matrix. I would also recommend authors to study around polymer/clay nanocomposites, where the significants of 2D nanosheets and exfoliation are often well described.

Comment 5.

Authors’ answer:

TG analysis of composites is a common method to measure under the conditions of inert atmosphere. Melt processing of plastic or composites is also performed under the inert atmosphere. When examining the degree of oxidation, it is common to perform TG anlaysis under oxygen conditions. If necessary, I will measure and present under oxygen conditions.

To be honest, I have not come across such an article, where TG analysis was performed to evaluate the stability of nanocomposite during melt processing, especially for polyolefin-based nanocomposites. I would recommend the authors to make the relevant citations.

Comment 6.

Authors’ answer:

The measurement method is corrected as follow and described in 3.5. Characterization. The chemical structures of the 2D-nanosheets powder were characterized via the transparent mode of Fourier transform infrared (FT-IR) spectroscopy.

The measurement condition is still not fully described. If the authors performed FT-IR for the nanocomposite in a film form, how to make the film and film thickness should be described. The most unclear point is that the fillers are powder, and it is impossible to be directly measured by transmission mode. I guess the authors utilized the KBr pellet method etc., such basic experimental information is indispensable for reproduction. I would recommend elaborating on it.

Comment 8.

Authors’ answer:

This part is corrected and explained in 3.5. as follows; X-ray diffraction (XRD) patterns of the nanosheets and catalysts powder were obtained on Phillips X-Pert PRO MRD diffractometer (Cu-Kα radiation) in a nitrogen atmosphere.

In general, the people in Ziegler-Natta catalyst field cover the XRD cell with thin film (Mylar film etc.), flame-seal a capillary, or use a specific inert cell. I recommend the authors to elaborate on the method more. Again, it is important for the reproduction of the experiments.

Comment 10.

Authors’ answer:

When evaluating the intercalation or exfoliation of hBN by XRD, it is not confirmed by shift of peak but by change of intensity of peak. An explanation of the intensity is given in 2.1. Characterization of Modified 2D-nanosheets and Catalysts as follows:

As shown in Figure 4, the reflection patterns of hBN. hBN-OH and hBN-OTES were observed at 2θ = 26.7°, 41.6°, 42.5°, 50.7°, 55.2° and 76.1° corresponding to the (002), (100), (101), (102), (004) and (110) planes, respectively. After oxidation of hBN, hBN-OH showed higher intensities for (002) and (004) peaks and a dramatically higher intensity ratio e.g. I002/I100 = 79.9 and I004/I100 = 3.5 as compared to hBN (I002/I100 = 23.7 and I004/I100 = 0.9), which might be attributed to the enhanced expore of the (002) basal plane because of the intercalation of few layered hBN-OH from hBN along the (002) plane. The reflection peaks also corroborated the high intercalation of hBN-OTES, the intensity ratios were I002/I100 = 126.9 and I004/I100 = 4.4, which is a significant increase compared to hBN-OH, indicating the introduction of OTES on the surface of hBN-OH.

Inserted in Revised manuscript, Line 103-111

I am afraid to say that I still have not been convinced by the explanation. If the crystallite size reduced, and it reached a few numbers of hBN sheets, the broadness of (002) diffraction peaks should be changed, which the fact is well represented by the Scherrer equation for example. The following article may help authors’ understanding [https://doi.org/10.1038/srep35532]. In any case, My conclusion from Figure 4 is that no exfoliation occurred at least in hBN-OTES, and it does not fit to Scheme 2. I recommend the authors to reconstruct the relevant discussion and conclusion.

Author Response

Reviewer 1.

Thanks for your detailed review of this manuscript.

Comment 2.

I misunderstood your comment 2. The excellent characteristics of 2D-nanosheets are listed, and the recent article of composites using it is briefly summarized and added to Introduction part as below. (Revised manuscript, Line 25-31)

Layered two-dimensional (2D) nanosheet-based polymer composites have been considerable interest in the polymer industry due to their multi-functional properties including low density, high aspect ratio and specific surface area, excellent electrical/thermal conductivity, gas barrier properties, high permittivity, excellent mechanical properties including toughness and compressive strength, etc.[1-4] The graphene-based polyolefin nanocomposites are the most studied the field of nanosheet-based polymer nanocomposites, owing to their significant potential and excellent physical properties [5-7].

Comment 5.

There are many articles that measure the thermal stability of polyolefin or polyolefin-based composites in the nitrogen or argon atmosphere in the analysis with TGA. Recent articles related to this is shown below:

Polym. Adv. Technol. 2020, 31, 1099-1109, Polymers 2020, 12, 597, Polymer 2020, 196, 122463, Mater. Today Commun. 2020, 23, 100880, J. Appl. Polym. Sci. 2020, 137, 48553, Polym. Compos. 2018, 39, 1361-1369, Compos. Part A 2017, 97, 120-127, Carbon 2016, 108, 274-282, Polym. Deg. Stab. 2009, 94, 39-48, etc.

Among them, 4 articles were cited in Reference No. 32 to 35. (Revised manuscript, Line 292, and Line 402-414)

Comment 6.

The measurement method is corrected as follow and described in 3.5. Characterization.

The chemical structures of the 2D-nanosheets were characterized via the transparent mode of Fourier transform infrared (FT-IR) spectroscopy (Jasco 4100, Japan). The spectra of samples were performed in the wavenumber range of 4000 ~ 500 cm-1 at 128 scans per and 4 cm-1 resolution using KBr pellet method.

(Inserted in Revised manuscript, Line 272-275)

Comment 8.

This part is corrected and explained in 3.5. as follows.

  X-ray diffraction (XRD) patterns of the nanosheets and catalysts were obtained on Phillips X-Pert PRO MRD diffractometer (Cu-Kα radiation). In the case of catalysts, the amorphous cover glass was covered to avoid oxygen and the 2Ɵ scanning range was 5~80º with a rate of 4º/min.

(Inserted in Revised manuscript, Line 275-278)

Comment 10.

As reviewer pointed out, the results of Figure 4 and the expression of Scheme 2 are not to be inconsistent. Even in this study, hBN was not fully exfoliated. So even XRD results in this study, the term ‘exfoliation’ was not used. As with FR-IR measurement, the XRD pattern analysis confirmed that OTES was combined in edge of hBN.

BN is known to be difficult to fully exfoliate with the chemical exfoliation method. The XRD analysis of this study also confirmed that there was little change in crystalline of hBN, and that it was a chemical modification due to the difference in intensity ratio. These results are explained in many papers besides Reference 23 to 25.

The hBN used in this study used few layered hBN as a liquid-assisted mechanical exfoliation method. Therefore, it is thought that there was little shift of the diffraction peak in the XRD pattern because it did not become a fully exfoliation. This method (a liquid-assisted mechanical exfoliation method) is described in addition to 3.2. Synthesis of MoS2-ODA and hBN-OTES. Then, in Scheme 2, it was modified by few layered hBN.

In the text, the expression ‘Exfoliation’ was not used, it is expressed as ‘prevention of aggregation’.

The correction is as follow;

As shown in Figure 4, the reflection patterns of hBN. hBN-OH and hBN-OTES were observed at 2θ = 26.7°, 41.6°, 42.5°, 50.7°, 55.2° and 76.1° which corresponded to the (002), (100), (101), (102), (004) and (110) planes of hBN skeleton. It indicates that the modification of hBN did not affect the crystalline structure of the hBN-OTES [23].

(Inserted in Revised manuscript, Line 105-107)

Few layered hBN was prepared from the bulk hBN via a liquid-assisted mechanical exfoliation method as follows; [30,31] 2g of the bulk hBN powder was added to a solution of 100 mL isopropanol and 100 mL deionized water. The mixture was mechanically stirred and simultaneously sonicated using ultrasonic bath at 80 W for 10h. The dispersions were centrifused to remove stacked hBN powders. The few layered hBN in the resultant supernatant was collected by centrifugation and then dried at 60 °C for 24h. The mass of few layered hBN was approximately 0.25g.

(Inserted in Revised manuscript, Line 238-243)

Scheme 2. Synthesis of hBN-OTES

(Inserted in Revised manuscript, Line 253-255)

MoS2 and hBN were modified with ODA and octyltriethoxysilane, respectively, for prevention of re-aggregation and good compatibility with PP.

(Inserted in Revised manuscript, Line 67-68)

Round 3

Reviewer 1 Report

Thank you very much for your sincere responses to my suggestions. The manuscript was much improved. I have no further comments. I believe the manuscript can be accepted.